# BAIV: An Efficient Blockchain-Based Anonymous Authentication and Integrity Preservation Scheme for Secure Communication in VANETs

Azees Maria [1], Arun Sekar Rajasekaran [1,*], Fadi Al-Turjman [2], Chadi Altrjman [2,3] and Leonardo Mostarda [4]

1   Department of ECE, GMR Institute of Technology, Srikakulam 532127, Andhra Pradesh, India;
    azees.m@gmrit.edu.in
2   Artificial Intelligence Engineering Department, Research Center for AI and IoT, AI and Robotics Institute,
    Near East University, Mersin 99138, Turkey; fadi.alturjman@neu.edu.tr (F.A.-T.); cmfaltrj@uwaterloo.ca (C.A.)
3   Chemical Engineering Department, University of Waterloo, Waterloo, ON N2L 3G1, Canada
4   Computer Science Division, Camerino University, 62032 Camerino, Italy; leonardo.mostarda@unicam.it
*   Correspondence: arunsekar.r@gmrit.edu.in

**Abstract:** Recent development in intelligent transport systems (ITS) has led to the improvement of driving experience in vehicular ad-hoc network (VANET) systems. Providing a low computational cost with high serving capability, however, is a critical phenomenon in the current VANET system. In the existing scenario, when the authenticated vehicle user moves from one roadside unit (RSU) to another RSU region, re-authentication of the vehicle user is required by the current RSU, which increases the computational complexity. To overcome the above-mentioned challenge, a blockchain-based authentication protocol is developed in this work. In this suggested process, blockchain is integrated with VANET, which enables the authentication of the vehicle user without the involvement of a trusted authority. Moreover, the integrity of the message and privacy of vehicle users are preserved in the blockchain network. Even though many blockchain-based schemes have been proposed recently, the existing schemes were not focused on conditional anonymity. However, in our proposed scheme, conditional privacy is introduced to revoke the malicious vehicles in the case of disputes and to avoid further damage to the VANET system. As a result, the proposed scheme provides an efficient mechanism for anonymous authentication, privacy, and integrity preservation with conditional tracking. Finally, the defense against different security threats is explained in the security analysis section, and the performance investigation section shows the competence and efficacy of our method with similar related methods.

**Keywords:** authentication; blockchain; integrity; security; revocation

## 1. Introduction

In the current digital age, human life is made very comfortable with the help of ITS. ITS plays a major role in reducing traffic complications and increasing traffic efficacy. Moreover, the services related to traffic rules and regulations and congestion of traffic are conveyed to the users well in advance, providing a better service for the users. In the current scenario, vehicles are connected to information technology, which helps to increase the safety of the user. Moreover, several features, like the proximity of emergency centers, hospitals, police stations, driving weather conditions, and so on, are to be shared with the nearby vehicles for efficient communication. With the introduction of the VANET system, the sharing of critical information has been made possible [1]. VANETs are distributed, self-organizing communication networks comprised of moving vehicles. The development of VANETs brings an incredibly comfortable and convenient driving experience for vehicle drivers. VANET has received significant attention for improving traffic safety and efficiency. Although it provides extensive information regarding safety along with infotainment services,

anonymous authentication and message integrity are still present practical concerns for the deployment of VANETs. Anonymous authentication and message integrity are needed to preserve the privacy of the user as well as to ascertain the legitimacy of the message. Two types of information exchange take place in the VANET system, namely, vehicle to roadside unit (V2R) communication, which takes place through a wired medium, and vehicle to another vehicle communication, which takes place through an open wireless medium [2].

The key component in intelligent vehicles is the onboard unit (OBU). The OBU is designed in such a way that it has high computational efficiency and is capable of generating the required keys. Every intelligent vehicle is fitted with an OBU, and each OBU communicates with the OBU of the nearby vehicle through a wireless medium. However, there are several security threats when the information transfer takes place through the open wireless medium [3]. As a result, there may be a possibility of information being hacked and modified. Therefore, securely transferring critical information to the vehicles plays a vital role [4]. Moreover, 5G communication plays a crucial role in the development of the VANET system [5]. If the security measures are not carried out, then the VANET system is completely affected by several security threats such as impersonation attacks, bogus message attacks, etc. Therefore, several security measures such as anonymous authentication, privacy, integrity preservation, and non-repudiation are to be implemented [6–8]. If authentication is not provided, there is a possibility of malicious vehicle users entering the system and performing an impersonation attack. To preserve the originality of the information transferred, the integrity of the message should be preserved. Moreover, the privacy of the authenticated user/RSU should be preserved anonymously [9,10]. In addition, if the real user turns out to be a malevolent user, his fake identity should be revealed by the trusted authority based on the conditional revoking mechanism.

The security problems in the VANET system are described in several related works. Most of the research focuses on the public key infrastructure (PKI) protocol. Here, two keys, namely public and private keys, are used to enhance security. The public key is generated from the corresponding private key, and it is computationally difficult to generate due to the discrete logarithmic problem (DLP). Two types of cryptosystems that are generally used for providing security are the RSA algorithm and ECC (elliptic curve cryptography). In the current technology, ECC is mostly used due to its smaller key size, which helps to increase the performance of the system.

In the modern existing authentication process, the vehicles and RSU should be registered with a trusted authority. If the vehicle user moves from one RSU region to another, then re-authentication of the vehicle user is mandatory. This involves more computational time and degrades the performance of the network. Further, there may be a possibility of several security threats. To overcome all these issues, blockchain is integrated with the VANET system in our suggested method, as it provides a high level of security. The main key features of blockchain technology include immutability, decentralization, distributed ledgers, consistency, security, integrity, and transparency [11–14]. The main contributions of this proposed method are as follows:

- To propose anonymous authentication to check the legitimacy of vehicles and RSUs.
- To propose a blockchain-based handover authentication for the vehicles in case of roaming.
- To propose conditional privacy to revoke the disobedient vehicles at any time.
- To propose integrity preservation to guard the communicating messages from the modification attack.

In recent years, many anonymous authentication schemes have been proposed based on signature generation to address most of the security and privacy concerns in VANET. However, due to the limitation of bandwidth and computation power, the efficiency of these schemes is questionable. In this work, we propose a scheme to provide anonymous authentication to the user and to achieve message integrity with less computation power by using bilinear pairing of points on the elliptic curve and integration with blockchain. The elliptic curve is used because it provides very strong security and it is not easy to

break. Digital signatures and hashing are used to authenticate the legitimate user, and the hash function is used to check the message's integrity. The digital signature is the process through which information is encrypted using the sender's public key and decrypted using the sender's private key, by which authentication is achieved because a digital signature can only be generated and cannot be regenerated. By using the hash function, the variable-length message is converted into a fixed-length message. In this proposed scheme, the legitimate OBU generates the hash function for the message that it wants to broadcast, and it sends the message as well as the hash function to other OBUs within its communication range. The receiving OBUs generate the hash function for the received messages and compare both the hash functions. The message is only accepted if both the values are matched, otherwise, the message is discarded.

This paper is structured as follows: The related works are briefed in Section 2. The system architecture is enlightened in Section 3, which deals with the system model, blockchain, and attack models. The proposed method is elucidated in Section 4, which deals with the anonymous authentication of both vehicle users and RSU. Moreover, efficient secure message transmission with integrity preservation and revocation is explained in this section. Section 5 elucidates the resistance of the suggested process against different possible attacks. The performance investigation section is described in Section 6. Lastly, the conclusion is drawn in Section 7.

## 2. Related Works

To enhance the security of VANETs, several methods have been proposed. Most of these methods focus on privacy and authentication, but the computational cost of all these options is considerably high due to subsequent authentication of the vehicle user when they move from one RSU region to another. In 2016, Mayank Satya Prakash Sharma et al. [15] suggested a communication model for VANET systems. This work practically describes the movement of vehicles in the VANET system under various cases. Safety messages are populated on the network frequently. The work is carried out using MATLAB software. In 2017, Deeksha et al. [16] proposed a solution that deals with security issues and their countermeasures in VANETs. This work briefly surveys the security threats and prevention procedures to be adopted. A comparative analysis of different attacks is focused on in this work. In 2018, Zhaojun Lu et al. [17] proposed a method focused on anonymous reputation using blockchain. This method focused on breaking the link between true identities. However, while this scheme preserves the privacy of the vehicle user, the communication cost and computational analysis are very high in this work. In 2018, Canhuang Dai et al. [18] proposed a new protocol for onboard units fixed to each vehicle. A new learning algorithm is used in this work, which describes whether to follow OBU requests or not. Based on prior knowledge, an innovative hotbooting procedure has been developed for the OBU. This will be useful for increasing the speed of operation. The utility ratio of OBU is enhanced using this scheme. However, this scheme does not address the computation and communication costs.

In 2019, Zhaojun Lu et al. [19] suggested a procedure based on privacy preservation using blockchain. Since blockchain is immutable and non-tamperable, the data recorded on the blockchain are highly secured. In this process, multiple certificates are used to achieve privacy for the vehicle user. In the event of any dispute, certificates from the blockchain are revealed. The computational analysis of this work is high due to the distributed authentication scheme. In 2020, Qi Feng et al. [20] suggested a scheme based on blockchain technology associated with the VANET system. The process is highly scalable and provides automatic authentication for vehicle users. If any malicious users are found in the network, they will be revoked by the trusted authority. Though security is enhanced in this work, it is vulnerable to reply attacks. In 2020, Djamel-Eddine Kouicem et al. [21] proposed a method for data sharing in VANETs. In this work, consortium blockchain is used for data sharing schemes. Even though the security is enhanced due to the incorporation of blockchain, the

storage cost of this scheme is high. Moreover, the efficiency of this method is reasonably low due to its high data sharing capability.

In 2020, Zhuo Ma et al. [22] proposed a procedure for a key management scheme in VANET. This key management protocol is implemented with the help of blockchain technology. This scheme can withstand several security threats such as tampering attacks, resistance to collision, and DoS attacks. The authentication scheme is built on a bivariate polynomial. Though this scheme is resistant to several well-known attacks, it suffers from considerable latency. In 2020, Bohan Li et al. [23] suggested an option for preserving the location of the vehicle user using blockchain. A new trust-based algorithm is designed in this work to enhance the security of data. Though the work focused on location privacy, the communication and computational analysis of this scheme are high. In 2020, Chao Lin et al. [24] suggested an option based on conditional privacy for VANETs. Certificate management is effectively achieved in this work. Every vehicle user in the VANET is provided with a certificate. Due to a large amount of certificate generation and verification, there is a high computational cost in this option.

In 2020, Gina El-Salakawy et al. [25] focused on data management using blockchain in VANETs. The messages are exchanged periodically among the different vehicles in the VANET system. Since blockchain technology is used in this scheme, the authorized messages are non-tamperable and immutable. The size of the blocks for the blockchain used in this work is high. In 2020, Jiao Liu et al. [26] suggested a scheme based on unlinkable authentication for the VANET system. In this work, the service manager acts as a node, and they are responsible for the mining process. The validity of the pseudonym is verified by the service manager. The work suffers from latency and storage costs. Huijie Yang et al. [27] suggested a privacy-preserving scheme in VANET based on cloud computing. Problems related to security can be solved using machine techniques. However, there is no efficient revoking mechanism for malicious vehicles. Farha Jahan et al. [28] researched the security aspects of autonomous systems. The development of the VANET system helps the vehicles to analyze traffic and avoid collisions. Moreover, the location-based information and the speed of the vehicles are analyzed in this work. An appropriate decision can be achieved based on the analysis of this work by the vehicle user. Zhenchang Xia et al. [29] reviewed the different challenges and key skills in VANETs. Data communication and transmission protocols are discussed in this work. Moreover, several security issues, reliability, etc., are focused on in this study. George Hatzivasilis et al. [30] suggested an efficient framework of resource sharing for VANET based on ambient intelligence. Moreover, in this work, the secure sharing of resources is achieved through a special scheme called mobile trust. However, there is no efficient anonymous authentication in this work. Yunhao Bai et al. [31] suggested real-time safety driving measures for the vehicle-to-vehicle communication. Here, the multi-channel protocol is suggested to enhance the security of the VANET system. However, the practical implementation of real-time communication is not conceivable. Teng Liu et al. [32] focuses on the user equilibrium state in VANETs. Flow control of the traffic is greatly reduced with the data caching approach. Moreover, the behavior of the vehicles, i.e., the traveling pattern, is analyzed in this work. Hani Sami et al. [33] suggested microservices for storing a large amount of data. The onboard unit (OBU) of the vehicle is a resource constraint and requires high processors for efficiency. The growing demand replaces this OBU with several micro-services on on-demand fogs. However, there is no efficient mechanism for the storage of large amounts of transmitted data in this work. Saadallah Kassir et al. [34] studied different architectures based on vehicle-to-vehicle (V2V) and vehicle-to-roadside unit (V2I). The link used between V2V and V2I is focused on in this work. Moreover, the performance and modeling of VANET and throughput are focused on in this work.

## 3. System Architecture

The System Architecture describes the overall structure of the VANET system, the communication of information between the entities in the VANET system, and also the

linkage of blockchain with the VANET architecture. The following section briefly explains the system model and blockchain in VANET.

### 3.1. System Model

The system model of VANET mainly entails three major entities: Trusted Authority, Roadside Unit, and Onboard Unit. The system model of the proposed system is shown in Figure 1. Blockchain is an independent network. In our proposed scheme, all the RSUs and TA are connected to the blockchain network independently. After successful anonymous authentication of each vehicle user, that authenticated information is stored in the blockchain network. Therefore, there is no need for re-authentication by consecutive RSU, as the data are stored in the blockchain. Thus, the RSU, the TA, and Blockchain are interconnected in Figure 1.

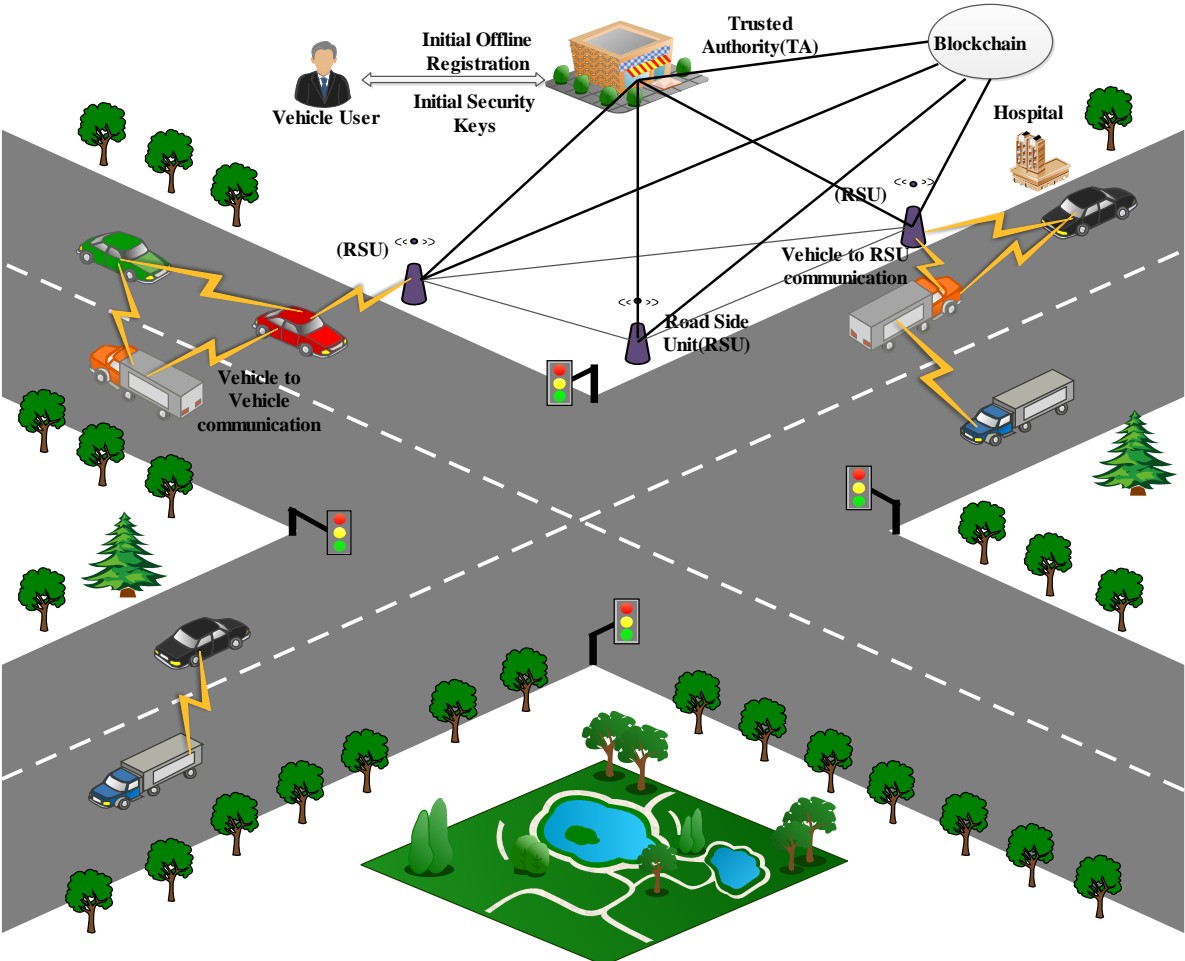

**Figure 1.** Proposed system model for VANET.

### 3.1.1. Trusted Authority (*TA*)

*TA* is the superior unit of the VANET system. It is responsible for the supervision of the complete VANET system, such as the registration of roadside units (RSU), the registration of onboard units (OBU), and the vehicle users, assigning them a unique registration ID. It has to manage and preserve the integrity of the database and software information. The vehicle user must satisfy all the requirements to register in *TA*. Hence, *TA* has all the information about RSUs, OBUs, along with some personal information about the vehicle user. *TA* is completely responsible for providing secure transmission of information between different entities in the VANET system. Moreover, *TA* plays a crucial role in revoking the malicious vehicle user from the VANET system based on a conditional tracking mechanism. As per the system architecture, all the RSUs within the region of *TA* are connected to *TA* using the

wired medium. Communication between RSUs and *TA* occurs through wired cables. The *TA* is responsible for generating the private and public keys.

### 3.1.2. Road Side Units (RSU)

The computing device installed along the roadside or in a particular spot such as a parking lot or at crossroads/junctions is called an RSU. Its purpose is to give local connection to the vehicles within its range. RSUs are linked to one another and also to the *TA*. The *TA* keeps track of any concessions in RSU performance. The network devices implanted in RSUs use IEEE 802.11p technology for their short-range dedicated communication. Each RSU is connected with neighboring RSUs and with *TA* through the wired network and to the vehicle user through the wireless network. Moreover, it provides location-based information to authenticated vehicles. The *TA* provides the required credentials to RSU.

### 3.1.3. On Board Unit (OBU)

Every VANET vehicle is equipped with onboard units for intelligent communication. An on-board chip consists of a set of hardware components that are assembled and programmed to serve their purpose. An OBU is fixed on every intelligent vehicle for communicating with each other. Moreover, it is implanted with GPS, which provides latitude, longitude, and time-based data for each vehicle. In addition, data recorders are also implanted in the OBU, which helps to record the vehicle crash information, similar to the black box in the aircraft.

### 3.2. Blockchain

The group of blocks are linked together to form the blockchain. The block is the form of distributed ledger which is linked together. The transactions recorded in the block are immutable and non-tamperable. Every block in the blockchain is linked through the hash of the previous block. Any modification in the single block will affect the entire blockchain. Moreover, the data loaded into the blocks are completely transparent. The transactions and all the information are displayed in the form of the SHA256 hash code. There is no third party governing the blockchain, thus it is completely decentralized. The linkage of the blocks in the blockchain is represented in Figure 2.

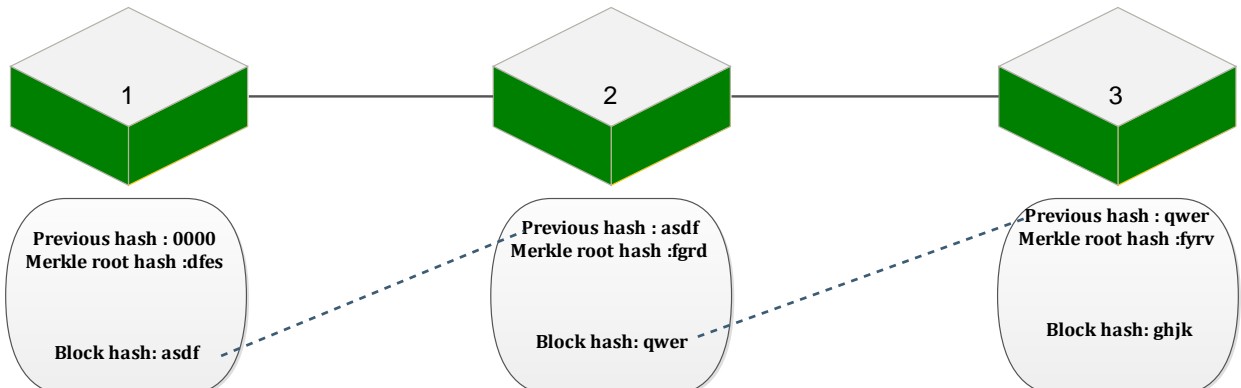

**Figure 2.** Linkage of blocks in blockchain network.

Blockchain in VANETs

When the vehicle user moves from one RSU region to another RSU region, there is a need for the vehicle user to be authenticated by the subsequent RSU. This will significantly increase the computational complexity and degrade the performance of the VANET system. Therefore, the introduction of blockchain technology has created opportunities for VANETs to resolve the above challenges. Thus, in our suggested framework, blockchain is integrated with the VANET system to enable authenticity without the involvement of *TA*, which reduces the computation time in VANETs. Initially, *TA* computes all the public and private

parameters and stores $D_{IDv}$, $A$ in the blockchain, where $A = e(P, Q)^{\gamma_i}$. The dummy variable is mapped to the original identity and is open to communicate with anyone. The RSU takes the dummy identity from the blockchain to create the authentication receipt, and the RSU transmits the authentication receipt to all the neighboring RSUs in order to avoid frequent re-authentication.

Once $A_{ID1}$ is received, the RSU computes $e(A_{ID1}P, V_{ID1})$ and checks this value in the blockchain. For that, $e(A_{ID1}P, V_{ID1})$ should be equal to $A$. Here, the blockchain is used to verify the authenticity without the involvement of *TA*. In connection to this, the re-authentication time will be reduced due to low computation time, which increases the system efficiency.

### 3.3. Attack Model

Internal attackers and external attackers are the two types of attackers in the VANET system. The attacker who performs malicious activities inside the VANET system is referred to as an "insider" or "internal attacker," whereas when the attack is performed from the external side and its influence has a great impact on the VANET system it is called an external attack. The proposed scheme should tolerate such forms of attacks. In the suggested work, the concentration is mainly focused on the external attacker. Some forms of different attacks are briefed as follows:

#### 3.3.1. Impersonation Attack

The adversary pretends to be a legitimate vehicle user or RSU and performs this attack. As a result, the sensitive information may be hacked.

#### 3.3.2. Fake Message Attack

The adversary sends a bogus message to the vehicle's user. As a result, the legitimate vehicle user believes the bogus message and performs the required function which leads to disaster.

#### 3.3.3. Privacy Revealing Attack

The privacy of the vehicle user or RSU is of principle importance. If the privacy of the entities is revealed, there is a possibility that sensitive information may be leaked.

#### 3.3.4. Masquerading Attack

The original login or password is hacked by the adversary and he performs the unauthorized access. As a result, there is a vulnerability to the leakage of secret information.

#### 3.3.5. Forgery Attack

The adversary forges the certificate/signature of the authenticated vehicle user/RSU and uses it to his advantage.

## 4. Proposed Work

In this work, an efficient blockchain-based anonymous authentication and integrity preservation for secure communication in VANETs is proposed. It entails seven sections, namely: system initialization, registration of a vehicle user, RSU registration, anonymous authentication of a vehicle user, anonymous authentication of an RSU, handover mechanism, integrity preservation, and, finally, revocation. The meanings of the symbols, variables, and parameters used in the proposed work are listed in the following Table 1.

### 4.1. System Initialization

The *TA* initially chooses the finite elliptic curve $y^2 = x^3 + ax + b \ mod \ q$, where $q$ is the large prime number. Moreover, $P$ and $Q$ represent the points of the finite elliptic curve. Then, the *TA* chooses random numbers $\alpha, \beta \in Z_q^*$, where $Z_q^*$ is the multiplicative group of size $q$ and computes its public key ($T_{pub}$) and verification key ($T_{ver}$) as $T_{pub} = \alpha P$

and $T_{ver} = \beta P$. In addition, the *TA* selects a hash function $H: \{0, 1\}^*$ and publishes $(T_{pub}, T_{ver}, H, P, Q, e(P, Q), q)$ to all users and RSUs in the VANET network.

**Table 1.** List of notations and abbreviations.

| Notations | Explanation |
|---|---|
| *TA* | Trusted authority |
| $y^2 = (x^3 + ax + b) \bmod q$ | Finite elliptic curve equation |
| *P* and *Q* | Points of the finite elliptic curve |
| $Z_q^*$ | Non-zero elements of a finite field $Z_q$, where $Z_q^* = [1, 2, \ldots, q-1]$ |
| $q$ | Large prime number |
| $\alpha, \beta, \gamma_i$ | Random numbers chosen from $Z_q^*$ by *TA* |
| $T_{pub}$ | Public key of *TA* |
| $T_{ver}$ | Verification key of *TA* |
| $H: \{0, 1\}^*$ | Secure hash function |
| $A_{ID1}$ | First authentication ID for vehicle user |
| $A_{ID2}$ | Second authentication ID for vehicle user |
| $D_{IDv}$ | Dummy identity for vehicle user |
| $V_{ID1}$ | First verification ID for RSU |
| $V_{ID2}$ | Second verification ID for RSU |
| $D_{IDR}$ | Dummy identity for RSU |
| $\theta_i$ | Signature of a message |
| $m_i$ | Original message |
| $m_i^*$ | False message |
| $t_i$ | Timestamp |
| $x_i$ | Random number chosen by RSU |
| $s_1, s_2$ | Short life keys of RSU |
| $u_i$ | Random number chosen by vehicle user |
| $\oplus$ | EXOR operation |

*4.2. Registration of Vehicles*

The registration of the vehicles is performed in the *TA*. At the time of vehicle registration, the vehicle users are required to submit their original credentials, such as phone number, personal ID, address, etc., to the *TA* directly. Then, the *TA* chooses $\gamma_i \in Z_q^*$ and computes the Authentication ID ($A_{ID1}$) as $A_{ID1} = \gamma_i(\alpha + \beta)$. After submitting the original identities to *TA*, the *TA* chooses a dummy vehicle ID as ($D_{IDv} \in Z_q^*$) for every user. A dummy identity is used to communicate with anyone. During the time of transfer of data, only dummy identity is exposed to other entities. The dummy identities are mapped with the real identities only in the *TA*. Even though the dummy identities are captured, it will provide zero knowledge about the original identities. This means that other users can anonymously authenticate the particular users in such a way that their privacy can be preserved, as well. Moreover, the *TA* computes the second authentication ID ($A_{ID2}$) as $A_{ID2} = H(D_{IDv} \times A_{ID1})$. Then, the *TA* returns ($\gamma_i, A_{ID1}, A_{ID2}$) to the vehicle users securely. In addition, the *TA* stores $D_{IDv}, A$ in the blockchain, where $A = e(P, Q)^{\gamma_i}$.

### 4.3. Registration of an RSU

The registration of RSUs is performed in the *TA*. The *TA* computes the verification ID ($V_{ID1}$) for each RSU as $V_{ID1} = \left(\frac{1}{\alpha+\beta}\right)Q$. Then, the *TA* chooses a dummy ID ($D_{IDR} \in Z_q^*$) for each RSU and computes the second verification ID ($V_{ID2}$) as $V_{ID2} = H(D_{IDR} \times V_{ID1})$. After successful registration, the *TA* secretly stores $V_{ID1}, V_{ID2}, \beta$ in the RSU.

### 4.4. Anonymous Authentication of a Vehicle User

Authentication is the process of verifying the credentials of the vehicle users to ensure security. In the anonymous authentication process, the authentication is done without revealing the real identities of vehicle users, hence maintaining the privacy of the users. Each OBU should perform anonymous authentication with RSUs and other OBUs in order to communicate with them. The following steps are executed:

1.  When the vehicle enters inside the RSU coverage region, the OBU of the vehicle sends $\gamma_i P$ to the RSU.
2.  Similarly, RSU will send $D_{IDR}P$ to the OBU.
3.  Then, the user will compute $k = \gamma_i D_{IDR} P$.
4.  Moreover, at the same time, the RSU computes $k = D_{IDR} \times \gamma_i P$.
5.  Later, the user computes $k_1 = A_{ID1} \oplus H(k)$ and sends $k_1$ to the RSU.
6.  By receiving $k_1$ from OBU, the RSU calculates $A_{ID1} = k_1 \oplus H(k)$. After getting $A_{ID1}$, the RSU computes $e(A_{ID1}P, V_{ID1})$ and checks this value in the blockchain. For that, $e(A_{ID1}P, V_{ID1})$ should be equal to $A$. Here, the blockchain is used to verify the authenticity without the involvement of *TA*. In connection to this, the re-authentication time will be reduced due to low computation time.

**Proof of correctness**

$$
\begin{aligned}
e(A_{ID1}P, \, V_{ID1}) &= \left(\gamma_i(\alpha+\beta)P, \left(\tfrac{1}{\alpha+\beta}\right)Q\right) \\
&= e(P,Q)^{\gamma_i(\alpha+\beta)/(\alpha+\beta)} \\
&= e(P,Q)^{\gamma_i} \\
&= A
\end{aligned}
$$

Then, the RSU takes the $D_{IDv}$ from the blockchain and creates the authentication receipt as $AR = (D_{IDR}, D_{IDv}, H(D_{IDR}, D_{IDv}))$. This receipt will be transmitted to all the upcoming RSUs to avoid frequent re-authentication. At the same time, the RSU calculates $k_2 = A_{ID1} \oplus D_{IDR}$. Then, the RSU sends $k_2$ to the user. By receiving it, the user extracts $D_{IDR}$ as $D_{IDR} = A_{ID1} \oplus k_2$.

### 4.5. Anonymous Authentication of an RSU

The RSU provides location-based information to the vehicles in its coverage region. Each VANET vehicle should authenticate the RSU to trust the information provided by the RSU. In this process, the RSU chooses $x_i \in Z_q^*$ and computes the following parameters:

1.  $u_i = x_i P$
2.  $\varphi_i = H\left(A_{ID1} \times T_{pub}\right)$
3.  $\lambda_i = (x_i + \varphi_i \beta) mod \, q$.

Then, it computes $s_1 = D_{IDR} \oplus \lambda_i$ and $s_2 = A_{ID2} \oplus \varphi_i$. After that, the RSU sends $u_i, s_1, \, s_2$ to the user. By receiving $(s_1, s_2, u_i)$, the user first retrieves $\lambda_i, \varphi_i$ and checks $\lambda_i P = (u_i + \varphi_i T_{ver})$. If this condition satisfies, the user accepts the RSU and gets location-based information.

**Proof of correctness**

$$
\lambda_i P = (x_i + \varphi_i \beta)P = (x_i P + \varphi_i \beta P) = u_i + \varphi_i T_{ver}
$$

*4.6. Secure Message Transmission and Integrity Preservation*

In order to send a message to another vehicle, the sender vehicle chooses $\mu_i \in Z_q^*$ and computes the following parameters:

1. $X_1 = \mu_i T_{pub}$
2. $Y_1 = \lambda_i T_{pub}$
3. $\wp = X_1 + Y_1$
4. Moreover, the sender vehicle chooses $a_i \in Z_q^*$ and computes $A_i = a_i T_{pub}$
5. $\eta_i = \delta_i(a_i + \mu_i + \lambda_i) \bmod q$, where $\delta_i = H(m_i \times A_i)$.

Then, the vehicle sets $\theta_i = (A_i, m_i)$ as the signature of a message. The integrity of the message will be preserved due to the unique nature of the signature which is attached to the message. Since the signature cannot be modified or altered by anyone, the integrity will be preserved. Then, the sender vehicle sends $(\eta_i, t_i, m_i, \theta_i, \wp, D_{IDv})$ to the other vehicle. Here, $t_i$ represents the timestamp at which the message is created. By receiving, the receiver vehicle computes $\delta_i = H(m_i \times A_i)$, and then it verifies whether $\eta_i T_{pub} = \delta_i(A_i + \wp)$. If it gratifies, the message $(m_i)$ is accepted, otherwise, it will be rejected.

**Proof of correctness**

$$
\begin{aligned}
\eta_i T_{pub} &= \delta_i(a_i + \mu_i + \lambda_i)T_{pub} \\
&= \delta_i\left(a_i T_{pub} + \mu_i T_{pub} + \lambda_i T_{pub}\right) \\
&= \delta_i(A_i + X_1 + Y_1) \\
&= \delta_i(A_i + \wp)
\end{aligned}
$$

*4.7. Revocation*

In our proposed scheme, if any vehicle user starts unusual behavior or attacks after having an authentication, the TA will revoke the particular malicious vehicle from the VANET system based on the complaints raised by the nearby vehicle users to the TA, as follows. After successful authentication, some vehicles may send false information to other vehicles for their own benefit. In this case, the *TA* revokes the misbehaving vehicles to prevent the misuse of the VANET system. As an example, let us consider that a false message $m_i^*$ is sent to the other vehicles, i.e., $(\eta_i, t_i, m_i^*, \theta_i, \wp, D_{IDv})$. By receiving this, the other vehicles may know that $m_i^*$ is false information. In such a case, $(t_i, m_i^*, \theta_i, D_{IDv})$ will be given to the *TA* through RSU. By seeing $(t_i, m_i^*, \theta_i, D_{IDv})$, the vehicle user with $D_{IDv}$ will be revoked and then *TA* sends $(D_{IDv}, H(D_{IDv}, \beta))$ to all the RSUs. By receiving this, RSU computes $F = H(D_{IDv}, \beta)$.

If $F$ is equal to received $H(D_{IDv}, \beta)$, then the $D_{IDv}$ will be stored in the block list of all RSUs. Hence, the vehicles with $D_{IDv}$ will not be allowed to make further communications.

Furthermore, Table 2 shows the complete workflow of the proposed work, and Table 3 portrays the flow diagram of anonymous authentication of an RSU and secure message transmission with other vehicle users, respectively.

**Table 2.** Proposed workflow diagram.

| Initialization Phase |
| :---: |
| Finite elliptic curve: $y^2 = x^3 + ax + b \bmod q$ |
| Points on curve: $P$, $Q$ |
| Random numbers $\alpha, \beta \in Z_q^*$ |
| Public key of *TA*: $T_{pub} = \alpha P$ |
| Verification key of *TA*: $T_{ver} = \beta P$ |
| Hash function: $H : \{0, 1\} \to Z_q^*$ |
| Public parameters: $(T_{pub}, T_{ver}, H, P, Q, e(P, Q), q)$ |

**Table 2.** *Cont.*

| Initialization Phase | |
|---|---|
| **Registration of vehicle user** | |
| *TA* chooses $\gamma_i \in Z_q^*$ | |
| $D_{IDv} \in Z_q^*$ | |
| $A_{ID1} = \gamma_i(\alpha + \beta)$ | |
| $A_{ID2} = H(D_{IDv} \times A_{ID1})$ | |
| $(\gamma_i, A_{ID1}, A_{ID2})$ ⟶ Vehicle user | |
| $(D_{IDv}, A)$ ⟶ Blockchain (where $A = e(P, Q)^{\gamma_i}$ | |
| **Registration of RSU** | |
| $D_{IDR} \in Z_q^*$ | |
| $V_{ID1} = \left(\frac{1}{\alpha+\beta}\right)Q$ | |
| $V_{ID2} = H(D_{IDR} \times V_{ID1})$ | |
| $(V_{ID1}, V_{ID2}, \beta)$ ⟶ *RSU* | |
| **Anonymous authentication of vehicle user** | |

| Vehicle user | RSU |
|---|---|
| $\gamma_i P, k = \gamma_i D_{IDR}P$ | $D_{IDR}P, k = D_{IDR} \times \gamma_i P$ |
| $k_1 = A_{ID1} \oplus H(k)$ | $A_{ID1} = k_1 \oplus H(k)$ |
| | Verifies $e(A_{ID1}P, V_{ID1}) = A$ |
| | $AR = (D_{IDR}, D_{IDv}, H(D_{IDR}, D_{IDv}))$ |
| $D_{IDR} = A_{ID1} \oplus k_2$ ⟵ | $k_2 = A_{ID1} \oplus D_{IDR}$ |

**Table 3.** Anonymous authentication of RSU and secure message transmission with integrity preservation.

| Anonymous Authentication of RSU | |
|---|---|
| **RSU** | **Vehicle User** |
| $x_i \in Z_q^*$ | - |
| $u_i = x_i P$ | - |
| $\varphi_i = H\left(A_{ID1} \times T_{pub}\right)$ | - |
| $\lambda_i = (x_i + \varphi_i \beta) \bmod q$ | - |
| $s_1 = D_{IDR} \oplus \lambda_i$ | - |
| $s_2 = A_{ID2} \oplus \varphi_i$ | - |
| $(s_1, s_2, u_i)$ ⟶ | $\lambda_i, \varphi_i$ |
| - | **Verifies** $\lambda_i P = (u_i + \varphi_i T_{ver})$ |
| **Handover authentication** | |
| **Vehicle user 1** | **Vehicle user 2** |
| $\mu_i, a_i \in Z_q^*$ | - |
| $A_i = a_i T_{pub}$ | - |

**Table 3.** *Cont.*

| Anonymous Authentication of RSU | |
|---|---|
| **RSU** | **Vehicle User** |
| $X_1 = \mu_i T_{pub}$ | - |
| $Y_1 = \lambda_i T_{pub}$ | - |
| $\wp = X_1 + Y_1$ | - |
| $\delta_i = H(m_i \times A_i)$ | - |
| $\eta_i = \delta_i(\ a_i + \mu_i + \lambda_i) mod\ q$ | - |
| $\theta_i = (A_i, m_i)$ | - |
| $(\eta_i, t_i, m_i,\ \theta_i, \wp, D_{IDv})$ $\longrightarrow$ | - |
| - | $\delta_i = H(m_i \times A_i)$ |
| - | **Verifies** $\eta_i T_{pub} = \delta_i(A_i + \wp)$ |

## 5. Security Analysis

The security analysis section briefly describes the resistance of the suggested scheme against different types of possible attacks.

### 5.1. Resistant to Impersonation Attack

To perform the impersonation attack, the adversary should pretend to be an authorized vehicle user/RSU in order to uncover the secret credentials. When the vehicle user enters into the RSU region, the authenticated vehicle user sends $\gamma_i P$ to RSU. Here, the value of $\gamma_i$ is chosen by the TA and it is transferred securely to an authenticated vehicle user in an offline way. Similarly, RSU will send $D_{IDR}P$ to the vehicle user. Here, the dummy identity $D_{IDR}$ of RSU is securely chosen by TA. Therefore, it is difficult for an adversary to compromise TA to obtain the required credentials. Thus, our suggested scheme provides defense against impersonation attack.

### 5.2. Resistant to Message Modification Attack

The message content should be changed to perform the message modification attack. The vehicle user sends $(\eta_i, t_i, m_i, \theta_i, \wp, D_{IDv})$ to the other vehicle. In order to send the fake message content, the message content $m_i$ should be changed. However, if the message content is changed, the end user not only checks the message content but also the value of $\eta_i$. This value is computed as $\eta_i = \delta_i(a_i + \mu_i + \lambda_i) mod\ q$, where $\delta_i = H(m_i \times A_i)$. Moreover, the value of $A_i$ is computed as $A_i = a_i T_{pub}$, and $a_i$ is the randomly chosen number by vehicle user as $a_i \in Z_q^*$. Therefore, the complexity of finding the random number involves the discrete log problem (DLP). It is therefore hard to find the value of $A_i$ and $\delta_i$. Thus, the value of $\eta_i$ sent by the authenticated vehicle user cannot be changed. Both the message $m_i$ and $\eta_i$ are thus interlinked and the suggested method provides defense against a message modification attack.

### 5.3. Resistant to Fake/Bogus Message Attack

The suggested protocol is resistant to bogus/fake messages either from RSU to vehicle user or from vehicle user to RSU. When the authenticated vehicle user sends $(\eta_i, t_i, m_i,\ \theta_i, \wp, D_{IDv})$ to another vehicle user, it is difficult for an adversary to find the values of $\eta_i,\ \theta_i, \wp, D_{IDv}$ and change the message format completely, since the dummy identity of the vehicle $D_{IDv}$ is provided by the *TA* securely to the vehicle user. The value of $\wp$ is calculated from $X_1$ and $Y_1$ parameters. To calculate $X_1$, the value of the randomly chosen number $\mu_i \in Z_q^*$ should be known. Further, in order to calculate $Y_1$, the value of $\lambda_i$ should be known, where $\lambda_i = (x_i + \varphi_i \beta)$. The value of $x_i$ is a randomly chosen number and $\varphi_i$

involves the hashed values of $H\left(A_{ID1} \times T_{pub}\right)$. Thus, it is difficult to find the randomly chosen numbers and hash value of the numbers, as this involves DLP.

### 5.4. Message Integrity and Unlinkability

The vehicle sets $\theta_i = (A_i, m_i)$ as the signature of a confidential message. Here, $A_i$ is calculated from the random number $a_i$, where $A_i = a_i T_{pub}$. It is hard to find the value of the random number. Moreover, the integrity of the message will be preserved due to the unique nature of the signature which is attached with the message. Since the signature cannot be modified or altered by anyone, the integrity will be preserved in our suggested scheme. Moreover, for each transferred message, a new signature is created. The creation of the signature involves $A_i$, which is comprised of a random number. Thus, there is a complete unlinkability between the successive messages.

### 5.5. Resistant to Reply Attack

To perform the reply attack, the fake message or modified message should be sent to the end entity within a stipulated time. However, in our suggested scheme, the timestamp is attached to each transferred message. The sender sends the message at a stipulated time interval to the end entity. If the adversary captures the message and modifies or changes the message completely, then the required message will not be delivered to the destination at the required time interval. If the received message is greater than the required specified time interval, then the end entity simply discards the received message. Thus, the suggested work provides defense against a reply attack.

### 5.6. Conditional Tracking

If the false message $m_i^*$ i.e., $\left(\eta_i, t_i, m_i^*, \theta_i, \wp, D_{IDv}\right)$ is sent to other vehicles, the other vehicles may know that $m_i^*$ is false information upon receipt. In that case, $\left(t_i, m_i^*, \theta_i, D_{IDv}\right)$ will be given to the *TA* through RSU. By seeing $\left(t_i, m_i^*, \theta_i, D_{IDv}\right)$, the vehicle user with $D_{IDv}$ will be revoked, and then *TA* sends $(D_{IDv}, H(D_{IDv}, \beta))$ to all the RSUs. By receiving this, RSU computes $F = H(D_{IDv}, \beta)$. If $F$ is equal to received $H(D_{IDv}, \beta)$, then the $D_{IDv}$ will be stored in the block list of all RSUs. Hence, the vehicles with $D_{IDv}$ will not be allowed to make further communication.

### 5.7. Conditional Privacy Preservation

During the anonymous authentication of a vehicle user or RSU, only the dummy identities are used. These dummy identities of vehicle users or RSUs are provided by *TA* during their initial offline registration. Moreover, during the data transfer, only dummy identities are used. Further, these dummy identities are mapped with the real identities in *TA*. Thus, if the dummy identity is captured by an adversary, he will have zero knowledge about the real identity. Thus, the privacy of the vehicle user/RSU is preserved in our suggested work.

### 5.8. Resistant to Non-Repudiation Attack

In our suggested scheme, the vehicle users or RSUs are registered in the *TA* through an offline manner. During initial offline registration, the required confidential original credentials are submitted by the entities to the *TA*. After successful authentication, *TA* provides the required credentials to the end entities. These credentials are used during anonymous authentication and message transfer. Therefore, the vehicle user or RSU cannot repudiate them.

## 6. Performance Analysis

The performance investigation is evaluated in terms of computational cost and RSU serving capability. The following section elucidates the analysis briefly.

### 6.1. Computational Cost

The execution period required for performing the cryptographic operations is referred to as computational cost. The critical cryptographic operations involved in our suggested method are one point addition, multiplication, hashing function, pairing operation, and XOR operation, respectively. The notations used for the above-mentioned operations are $Ex_a$, $Ex_m$, $Ex_h$, $Ex_p$, and $Ex_{xor}$. The computational cost investigation of our suggested scheme is compared with existing similar works such as the schemes of Azees et al. [35], X. Lin et al. [36], Zhang et al. [37], and R. Lu et al. [38]. The entire execution is carried out using the CYGWIN platform 1.7.35 [39] with the system requirements of a Core i7, 3.4 GHz processor, 8 GB memory, and GCC version 4.9.2. Cygwin platform based PBC library coding is used. The pairing based cryptographic library is used in this work [40]. The execution time for the accomplishment of the point addition operation is 0.011 ms (milliseconds). Similarly, the execution times for the accomplishment of multiplication, hashing function, pairing operation, and XOR operation are 2.4 ms, 0.01 ms, 2.9 ms, and 0.01 ms, respectively. Table 4 shows the computational investigation of different schemes with the proposed scheme. From the table, it is clear that for the single user and single RSU verification, in the Azees et al. scheme, two pairing operations and five-point multiplication operations are required. Therefore, the total computation time for the Azees et al. scheme is given as $2Ex_p + 5Ex_m = 17.8$ ms. In case of $n$ users and $n$ RSUs verification, this scheme requires $(1 + n)$ pairing operations and $5n$ point multiplication operations. Then, the total verification cost for $n$ users and $n$ RSUs is given as $(1 + n)Ex_p + 5nEx_m$. For the single-user and single RSU verification, as in the X. Lin et al. scheme, three pairing operations and nine-point multiplication operations are required. Therefore, the total computation time is given as $3Ex_p + 9Ex_m = 30.3$ ms. In case of $n$ users and $n$ RSUs verification, this scheme requires $3n$ pairing operations and $(3 + 6n)$ point multiplication operations.

**Table 4.** Authentication time of various schemes.

| S.no | Scheme | Single User and Single RSU Authentication (in ms) | $n$ User's and $n$ RSU's Authentication (in ms) |
|------|--------|--------------------------------------------------|-------------------------------------------------|
| 1. | Azees et al. | $2Ex_p + 5Ex_m = 17.8$ | $(1 + n)Ex_p + 5nEx_m$ |
| 2. | X. Lin et al. | $3Ex_p + 9Ex_m = 30.3$ | $3nEx_p + (3 + 6n)Ex_m$ |
| 3. | Zhang et al. | $3Ex_p + 4Ex_m + 3Ex_h = 18.33$ | $3nEx_p + (2n + 2)Ex_m + 3nEx_h$ |
| 4. | R. Lu et al. | $4Ex_p + 10Ex_m = 35.6$ | $(3 + n)Ex_p + (4 + 6n)Ex_m$ |
| 5. | Proposed scheme | $4Ex_m + Ex_p + Ex_a + 3Ex_{xor} = 12.54$ | $4nEx_m + nEx_p + nEx_a + 3nEx_{xor}$ |

Then, the total verification cost for $n$ users and $n$ RSUs is given as $3nEx_p + (3 + 6n)Ex_m$. For the single user and single RSU verification, in the Zhang et al. scheme, three pairing operations, four-point multiplication operations, and three hash function operations are required. Therefore, the total computation time for Zhang et al. scheme is given as $3Ex_p + 4Ex_m + 3Ex_h = 18.33$ ms. In case of $n$ users and $n$ RSUs verification, this scheme requires $3n$ pairing operations, $(2n + 2)$ point multiplication operations and $3n$ hash function operations. Thus, the total verification cost for $n$ users and $n$ RSUs is given as $3nEx_p + (2n + 2)Ex_m + 3nEx_h$. For the single user and single RSU verification, in the R. Lu et al. scheme, four pairing operations and ten-point multiplication operations are required. Therefore, the total computation time is $4Ex_p + 10Ex_m = 35.6$ ms. In case of $n$ users and $n$ RSUs verification, this scheme requires $(3 + n)$ pairing operations and $(4 + 6n)$ point multiplication operations. The total verification cost is therefore given as $(3 + n)Ex_p + (4 + 6n)Ex_m$.

For the single user and single RSU verification, in the proposed scheme, four pairing operations, a one-point multiplication operation, a one-point addition operation, and three XOR operations are required. Therefore, the total computation time for the proposed scheme is given as $4Ex_m + Ex_p + Ex_a + 3Ex_{xor} = 12.54$ ms. Therefore, in one second of

time, the RSU can authenticate approximately 80 vehicle users. Thus, in one minute, an RSU can authenticate nearly 4800 vehicle users. Nevertheless, practically, such a large number of vehicle users cannot cross the particular RSU region in a single minute. Even so, if that were possible, our proposed scheme is computationally efficient to authenticate 4800 vehicle users/minute.

The simulation results for vehicle user authentication and RSU authentication are shown in Figures 3 and 4, respectively. Figures 5 and 6 show the message transmission results and computational values for different schemes.

**Figure 3.** Simulation result of vehicle user authentication.

**Figure 4.** Simulation result of RSU authentication.

**Figure 5.** Simulation result of message transmission.

```
\ Execution Time \
tm = 0.002498
tad = 0.000011
th = 0.000001
tp = 0.002908
total computation time for 20 message prs: 258.279999
total computation time for 20 message az: 310.899994
total computation time for 20 message lin: 484.000000
total computation time for 20 message zh: 279.000000
total computation time for 20 message lu: 376.700012
total computation time for 40 message prs: 516.559998
total computation time for 40 message az: 618.900024
total computation time for 40 message lin: 958.000000
total computation time for 40 message zh: 553.000000
total computation time for 40 message lu: 734.700012
total computation time for 60 message prs: 774.840027
total computation time for 60 message az: 926.900024
total computation time for 60 message lin: 1432.000000
total computation time for 60 message zh: 827.000000
total computation time for 60 message lu: 1092.699951
total computation time for 80 message prs: 1033.119995
total computation time for 80 message az: 1234.900024
total computation time for 80 message lin: 1906.000000
total computation time for 80 message zh: 1101.000000
total computation time for 80 message lu: 1450.699951
total computation time for 100 message prs: 1291.400024
total computation time for 100 message az: 1542.900024
total computation time for 100 message lin: 2380.000000
total computation time for 100 message zh: 1375.000000
total computation time for 100 message lu: 1808.699951
```

**Figure 6.** Simulation results for the computational cost of different schemes.

In this VANET system, each *TA* is responsible for a particular region. This region is separated into several subdomains controlled by RSUs. Since the range for wireless communication is limited, separating the RSU to the larger distance is practically impossible. In relation to this, even though the deployment cost of the RSUs is greater, for the efficient functioning of the VANET system, the RSU should be placed every 300 m. Several RSUs are interconnected within a short range of 300 m through the wireless communication network. In case of *n* users and *n* RSUs verification, this scheme requires 4*n* pairing operations, *n* point multiplication operations, *n* point addition operations, and 3*n* XOR operations. In connection to this, as shown in Table 4, the total verification cost for *n* users and *n* RSUs is given as $4nEx_m + nEx_p + nEx_a + 3nEx_{xor}$.

The graphical view of the computational cost is shown in Figure 7. In our proposed scheme, whenever the vehicle enters the VANET system, the vehicle user is authenticated by the first RSU, and its authenticated information is updated on the blockchain network. During the movement of the vehicle user from one RSU region to another subsequent RSU region, there is no need for the new RSU to re-authenticate the vehicle user. Instead, the subsequent RSU will take the required authenticated data of the vehicle user from the blockchain. Thus, the computational cost is significantly reduced. For instance, for authenticating 100 vehicle users, our scheme requires 1291 ms, whereas the other existing schemes consume more than 1375 ms for authenticating the same number of users, as shown in Figure 7. From the above-obtained graph, we can infer that the previously existing related schemes consume more computational time, while the proposed scheme takes much less time to carry out these operations, and it is proven to be computationally efficient. For all other schemes mentioned, there are more pairing operations required, whereas the suggested work requires only one pairing operation, a four-point multiplication operation, a one-point addition operation, and three XOR operations for authenticating a single-vehicle user and RSU. If the computational time is greater, then the time taken to carry out a specific cryptographic operation is also greater. To obtain the full speed performance characteristics, the computational time of a cryptographic operation should be sufficiently diminished. If the computation time is shorter, then the scheme is more preferable and the performance is also higher relative to the other schemes. In our proposed schemes, the time

period for the cryptographic operations is computed based on 100 random simulations, and the average value is taken for the calculations. Moreover, VANET is a huge network system that consists of a large number of vehicles. Thus, the computational cost is calculated for 100 users. If the number of cars is large, only initial authentication is performed for each car by RSU. After that, the information is stored in the blockchain and the subsequent RSUs take the required information from the blockchain network. As a result, there is no re-authentication of vehicle users. Our scheme consumes only 12.54 ms for authenticating a single user. Therefore, in one second, our scheme can authenticate approximately 80 users. However, practically, 80 vehicle users cannot cross the particular RSU in one second. Our scheme, therefore, is computationally efficient for a real-time environment. Moreover, in V2V or V2R communication, the data transmission happens in an anonymous way, and so, the communication overhead will not have much more impact on the security of the proposed scheme, irrespective of the number of bits.

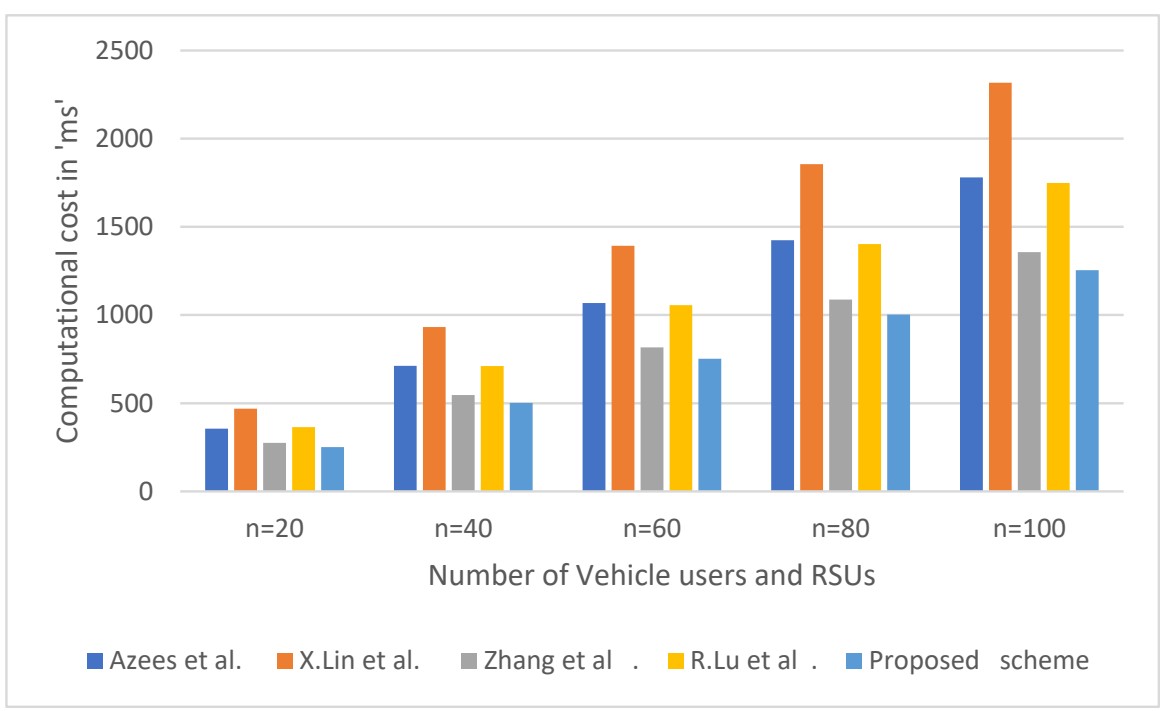

**Figure 7.** Graphical representation of computational cost.

### 6.2. RSU Service Providing Capability

RSU serving capability denotes the successful service provided by the RSU to the vehicle user in its coverage region. When the vehicle user is present in the RSU coverage area, the required location-based information should be conveyed to the authenticated vehicle user effectively. Let $\mathbb{N}$ be the number of authenticated vehicle users in the coverage region and $p$ be the probability of the service provided by RSU to the $\mathbb{N}$ users. The authentication time for providing service to the single user is computed as $\Delta = 4Ex_m + Ex_p + Ex_a + 3Ex_{xor} = 12.54$ ms. Therefore, the total serving capability of RSU is computed as $RSU_{ser} = \frac{p}{\mathbb{N} * \Delta * \mathbb{N}}$. Figure 8 represents the serving capability of RSU. Figure 8 clearly indicates that, as the number of vehicle users increases, the computation time also increases, with a decrease in the RSU serving capability ratio.

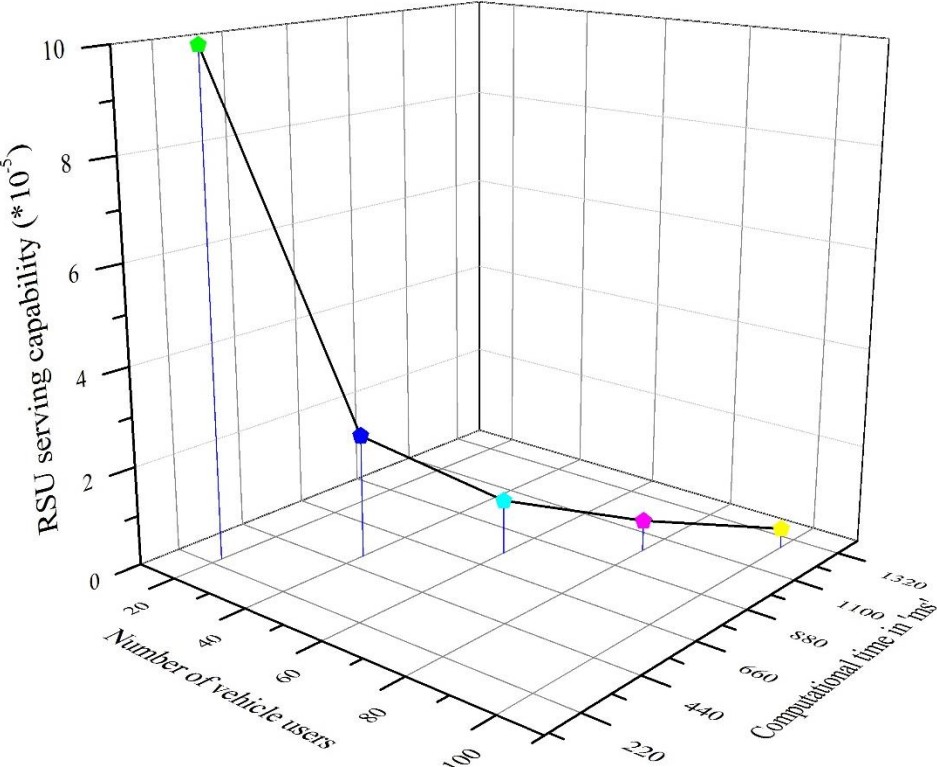

**Figure 8.** RSU serving capability.

## 7. Conclusions

In this work, an initially anonymous authentication of vehicle users is proposed. Here, the blockchain is used to provide the authenticity for the vehicle user without the involvement of a trusted authority, and re-authentication by the subsequent RSU is completely evaded. To trust the location-based information received from the RSU, the vehicle user performs the anonymous authentication of RSU. Moreover, the message is transferred from one vehicle user to another by an efficient handover mechanism. The signature and timestamp help to preserve the integrity and are resistant against a reply attack. In addition, the suggested scheme is tolerant against several possible attacks, such as non-repudiation, impersonation, bogus/fake message, message alteration attacks, and so on. Finally, the performance of the suggested work is evaluated in terms of computational cost and RSU serving capability. Future work can be extended to the inclusion of batch authentication integrated with artificial intelligence for attaining low computational cost during authentication. In addition, security issues related to 6G and edge computing are to be incorporated. Moreover, a lightweight revocation scheme based on elliptic curve cryptography will be adopted. Further, an identity based verification scheme for a group of vehicles, message delay, and message transmission loss are to be focused in future work.

**Author Contributions:** A.M., proposed work, security analysis, paper writing. A.S.R., experimental work, literature survey, paper writing. F.A.-T., performance analysis, system model. C.A., literaturesurvey, computational cost, paper writing. L.M., proposed work, system model. All authors have read and agreed to the published version of the manuscript.

**Funding:** The authors declare no funding received for this research.

**Data Availability Statement:** Not applicable.

**Conflicts of Interest:** The authors declare that they have no conflict of interest regarding the publication of this paper.

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
