# Peer review of "BAIV: An Efficient Blockchain-Based Anonymous Authentication and Integrity Preservation Scheme for Secure Communication in VANETs"

_electronics, doi:10.3390/electronics11030488_

Round 1
Reviewer 1 Report
A well-written paper that offers interesting insight on anonymous authentication. The figures included are very well created and help the reader gain a better understanding of the text (figures are well integrated in the text). Figure 2 appears to be missing a legend. The work carried out is explained in a methodical way as are the results. Perhaps figure 4 could be included before the conclusion section. Future steps are part of the conclusion which is an advantage. The references are relevant and up-to-date.
Reviewer 2 Report
This is a good paper that presents a blockchain-based authentication protocol for vehicular ad-hoc network (VANET) systems that enables the authenticity of the vehicle user without the involvement of trusted authority.
The manuscript is well written and organized. The theory seems correct and the reported comparisons are interesting.
However I suggest adding some pseudocodes or block diagrams to better illustrate the proposed general scheme.
Finally, in Table 1 the time (in ms) should be indicated for the different schemes.
Reviewer 3 Report
The work proposes a Blockchain-based anonymous authentication and integrity preservation scheme for secure communication in vehicular ad-hoc networks (Vanets) called BAIV. The paper starts with an introduction about the topic and provides a comprehensive state of the art section. The system architecture is presented in detail and has afterwards been evaluated in a performance analysis. The results look promising. The future work section should be more detailed with references to other works.
Reviewer 4 Report
This paper presents a blockchain-based authentication scheme in VANETs. Recently, there have been many similar studies, and the authors should describe why this scheme is essential in solving practical issues with specific example-use of vehicular communication.
The framework of VANETS for a realistic traffic network is missing. The basic purposes and implementation issues should be described well before introducing the proposed authentication scheme. Figure 1 is confusing since many RSUs are shown for a single intersection. How much length or traffic volume is covered by a single RSU? How do the vehicles switch, or are they allocated among RSUs? It is not feasible to introduce an RSU at every 100 m of a city road network with a total length of hundreds of km. The practical aspects of RSU based VANETs should be discussed.
Figure 1 also shows blockchain as an independent unit similar to RSU or TA, which is confusing. How can blockchain communicate, as shown by arrows, in a similar way to others? Is the blockchain located outside of these units as a database?
Recently, many blockchain-based schemes have been proposed for VANETs, with high computational costs due to subsequent authentication of the vehicle users. However, the proposed scheme provides only marginal differences in cost compared with them.
Authors describe various types of attacks. How the proposed scheme prevents such attacks is not explained clearly. What happens when a vehicle starts unusual behavior or attacks after having an authentication? Is TA monitoring such behavior each time step?
Why is a so simple block shown as a huge figure on page 6, with no caption?
Section 4.1 contains an equation. What are the meaning of the symbols, variables, and parameters used in the equation? All the symbols used must be explained at the proper place of their first use.
Figure 3 contains redundant information, as the bar values are obtained by multiplying the per-user cost with the number of users. Only the price per user is enough to understand the cost patterns. Others should be removed to avoid misleading by providing bar graphs only for n=1.
If the number of cars is large and the authentication process takes a long time, how much efficient communication is possible with the blockchain-based scheme? If the user number is high, e.g., n=100, the computation time exceeds 1 s. If authentication takes such a considerable time, how the vehicle can communicate with each other, which usually should be a few times per sec? Usually, in the standard V2V broadcast-based communication protocol approved by the authority of various countries, each vehicle broadcasts its 256 bits of information every 100 ms. Which is also safe from direct attacks or hacking, though one can broadcast misleading information. A proper discussion should be made to justify the proposed VANNET scheme with such a simple V2V communication, in terms of cost-benefits trade-off.
Round 2
Reviewer 2 Report
The Authors satisfied my suggestions and comments (in Table 4 it would have been better to also add the numerical values ​​in ms) and the manuscript was further improved.
Now, I can recommend to publish this manuscript in its current form.
Reviewer 4 Report
Although the authors have explained my raised earlier points, some of them are not clarified in the manuscript. The author should add some statements that are explained in the response letter.
